# PROGRAMMABLE NEURAL NETWORK TROJAN FOR PRE-TRAINED FEATURE EXTRACTOR

## ABSTRACT

Neural network (NN) trojaning attack is an emerging and important attack that can broadly damage the system deployed with NN models. Different from adversarial attacks, it hides malicious functionality in the weight parameters of NN models. Existing studies have explored NN trojaning attacks in some small datasets for specific domains, with limited numbers of fixed target classes. In this paper, we propose a more powerful trojaning attack method for large models, which outperforms existing studies in capability, generality, and stealthiness. First, the attack is programmable that the malicious misclassification target is not fixed and can be generated on demand even after the victim's deployment. Second, our trojaning attack is not limited in a small domain; one trojaned model on a large-scale dataset can affect applications of different domains that reuses its general features. Third, our trojan shows no biased behavior for different target classes, which makes it more difficult to defend.

## 1 INTRODUCTION

*Neural Network (NN) Trojaning Attack* or *Neural Network Backdoor Injection Attack* is an important attack model that can broadly damage the system based on NN models (Dumford & Scheirer, 2018; Liu et al., 2017; Liao et al., 2018; Gu et al., 2017). NN trojaning attacks hide malicious functionality inside the weights of an NN model, either by poisoning datasets or performing weight perturbation (Gu et al., 2017). The trojaned NN model predicts correct labels normally for legitimate inputs, and only misclassifies the inputs with trigger patterns to predefined target labels. The NN models are essentially just a set of weight parameters connected with certain network architectures. Their behavior highly depends on the weight parameters, but the meanings are completely implicit. Thus, modifying the weight parameters usually shows no difference to consumers. To note, it is a different attack model from adversarial attacks (Kurakin et al., 2016), which craft adversarial inputs to mislead NN models.

The NN trojaning attack is becoming an emerging practical and destructive attack model because of the broad usage of pre-trained models. Training a neural network with good features requires not only a large number of computing resources but also large-scale datasets. Thus, using pre-trained models is a common practice in developing NN-based applications to reuse expensive well-learned features. Accordingly, there are many open-source pre-trained models available online. They are produced by various companies, open-source communities, or personal maintainers, and consumed by end-users who may use these models directly or reuse part of them for a particular task. These pre-trained models benefit the agile deployment and boom the NN technique evolution. However, they also raise security issues since some vicious model promulgators can hide malicious functionalities in the clean model and release them for public use, which can be easily spread. Therefore, it is important to explore and understand the NN trojaning attacks.

Although the trojaning attack requires attackers to be capable of modifying the weight parameters of the NN model, it does not have to be an entire white-box. In terms of the attack scenarios, such trojaning attacks can be classified into two types, *outsourced training attack* and *transfer learning attack*. The first assumes that the victims will use the trojaned model directly without any further modification. This kind of attack is completely white-box and most existing studies focus on this assumption (Dumford & Scheirer, 2018; Liu et al., 2017; Liao et al., 2018). However, in real cases, victims often employ pre-trained NNs as well-learned feature extractors and further develop their

models (Gu et al., 2017). Therefore, the trojan attacks should resist victims' modifications, which is referred to as the *transfer learning attack* (Gu et al., 2017). One of such studies, BadNet (Gu et al., 2017), has explored the transfer learning attack in some small datasets on traffic signs.

Thus, although existing studies make initial steps that explore the potential effectiveness of trojan in transfer learning, their methodologies are restricted in small domains and validated on small datasets. For general feature extractors that are trained on large datasets and are used broadly, the attack is more challenging and the existing trojaning methods cannot be applied to this scenario: The victim tasks are completely unknown to the attackers and the target label that the attackers misleadingly train the trojaned model to recognize may even not be involved in the victim task.

For example, one of the official tutorials provided by TensorFlow[1] introduces the transfer learning scenario for image classifications. They use ImageNet (Deng et al., 2009) pre-trained models as well-learned image feature extractors and retrain the fully connected (FC) layers for new tasks on smaller Flower datasets. The official tutorial provided by MXNet[2] also introduce this scenario that transfer pre-trained VGG16 model for Caltech-256 dataset. In the natural language processing (NLP) field, it is also a popular practice to reuse BERT (Devlin et al., 2018) as pre-trained word-level features to solve many different kinds of NLP tasks. These scenarios are more realistic and trojans on those general features will affect a large scope of applications. Thus, it is important to explore the trojaning attack on the general feature extractors.

Another limitation of existing studies is that the trigger patterns of the trojans are usually handcrafted patterns for only one or few target classes. The limited diversity makes the trojans highly correlated with the trigger patterns. Defense methods (Chen et al., 2018; Liu et al., 2018a; Wang et al., 2019) based on statistic could detect or erase these trojans easily.

In this paper, we propose an NN trojaning attack method that is much more powerful, general, and stealthy. Instead of using a static set of handcrafted patterns to trigger a predefined target class, we use dynamic patterns to trigger any intended target class, which makes our trojan attack programmable. We can use a *target image* to describe the target class and generate a trigger pattern based on this image to encode and pass the information of the misclassification target to the trojan.

The dynamic trigger pattern makes our trojan much more powerful and general: Even if the explicit classes used in victim model are not involved in the pre-trained model and unknown to attackers, they can still describe the input they expect the victim model to see with a *target image* and then generate the corresponding pattern to trigger the malicious behavior. Further, the dynamic trigger also greatly increases the diversity of trigger patterns, which makes it more stealthy.

We demonstrate our attack method under the scenario described in the retraining tutorial from Tensorflow and MXNet, which uses pre-trained ImageNet (Deng et al., 2009) models and replaces the FC layers for the Flower dataset and Caltech-256 dataset. We insert a trojan into the ImageNet model and attack the victim model for the two smaller datasets. The trojan remains effective for both cases. Note that, the classes in the Flower dataset and Caltech-256 are not involved in the 1000 classes of ImageNet and attackers have no access to these datasets. The same trojaned model can affect victims using any other dataset.

## 2 RELATED WORK

Neural networks show vulnerabilities to the crafted adversarial inputs, which is referred to as *adversarial attack* (Kurakin et al., 2016). NN trojan is another important attack model which can broadly damage the systems based on NN models. In such an attack model, the NN model intellectual property (IP) vendors could be the potential attackers who hide malicious functionalities in the pre-trained NNs (Liu et al., 2017; 2018b; Wang et al., 2019). These models perform normally with legitimate inputs and can export targeted or untargeted outputs with the trigger inputs.

Previous studies have made some initial steps in the NN trojaning techniques. In most existing studies (Dumford & Scheirer, 2018; Liu et al., 2017; Liao et al., 2018), they assume that the victim

---

[1]*How to Retrain an Image Classifier for New Categories.* `https://www.tensorflow.org/hub/tutorials/image_retraining`

[2]*Fine-tune with Pre-trained Models.* `https://mxnet.incubator.apache.org/versions/master/faq/finetune.html`

Table 1: Comparison between our work and related work

| Capability | Transferability | Out-scope target | Dynamic target | Large Dataset |
|---|---|---|---|---|
| Dumford & Scheirer (2018) | × | × | × | × |
| Liu et al. (2017) | × | × | × | × |
| Liao et al. (2018) | × | × | × | × |
| Gu et al. (2017) | ✓ | × | × | × |
| Ours | ✓ | ✓ | ✓ | ✓ |

will adopt the pre-trained NN models directly, which is termed *outsourced training attack*. However, this situation rarely actually occurs. In practice, users typically fine-tune the FC layers of the pre-trained models to adapt to their working scenarios, which makes the attack more challenge; it is termed as *transfer learning attack*. Although the most related work, BadNet (Gu et al., 2017), has implemented a transfer learning attack, the triggers in their work are based on handcrafted patterns, which are statistically fixed. Therefore, their triggers can only support fixed target classes that are included in the pre-trained models. It cannot be applied to the scenario we demonstrate in this paper. Further, existing studies only demonstrated a high success rate of trojaning attack on small dataset such as MNIST (Dumford & Scheirer, 2018; Liao et al., 2018; Gu et al., 2017; Liu et al., 2017; Wang et al., 2019), face recognition (Dumford & Scheirer, 2018; Wang et al., 2019), traffic sign (Liao et al., 2018; Gu et al., 2017; Chen et al., 2018), and CIFAR10 (Chen et al., 2018). But people seldom use pre-trained models on these tiny datasets from an untrusted source. We compare our work with related studies in Table 1: We support target classes outside the pre-trained models, termed as *out-scope target*, and the target class is not fixed, termed as *dynamic target*. These properties make our attack much more powerful. We also demonstrate the attack on ImageNet.

There are also some initial studies about the defense of the NN trojan. Some detect if the dataset is poisoned (Chen et al., 2018), some detect if the model is poisoned by comparing the decision boundary of different classes (Wang et al., 2019), and some try to remove the trojan by squeezing the redundencies (Liu et al., 2018a). Most of them just work on trojaning attacks with just one or a few fixed target labels; in Section 5.3 we will analyze their effects on the proposed attack model.

## 3 THREAT MODEL

Figure 1 shows a typical flow of *transfer learning attack* for NN trojans (Gu et al., 2017). For the ease of understanding, we first explain several terminologies. The start of the flow is a pre-trained NN model, denoted as *clean model*; its task is *original task*. The network architecture of the clean model usually consists of a *backend model* and a *frontend model*. The backend model produces *general features* for a certain domain, which is intended to be reused by victims. The frontend model uses the general features for the underlying tasks and victims will develop their frontend model based on the backend model. The clean model usually comes from public model zoos or produced by attackers. Then, the threat model usually contains the following three phases.

**Trojaning Phase.** In this phase, attackers can fully access and make modifications to the entire clean model. They usually modify only the backend model to hide the trojan because the frontend model is replaced in later phases. The modified backend is denoted as *trojaned backend* and the entire model is now denoted as *trojaned model*. The trojaned model has the same network architecture as the clean model. The only difference is the weight parameters in the backend.

**Victim Phase.** The trojaned model is then distributed online and reused by victims. Victims intend to reuse the well-learned general features from the backend model for their new tasks, denoted as *victim task*. It is typically done by designing a new frontend, *victim frontend*. And the entire model now is denoted as *victim model*. Note that the victim frontend is unknown to attackers, including the explicit classes involved. On the other hand, although victims can fully access the trojaned model, they are unaware of the explicit method to trigger the malicious functionality.

**Trigger Phase.** The victim model is then deployed to real applications and the applications may also integrate other components. Now, victims are still capable of accessing the runtime information of their victim model and the system. But for attackers, it is a black box now except for the application scenario. Attackers can make small modifications to the input to trigger the malicious functionality

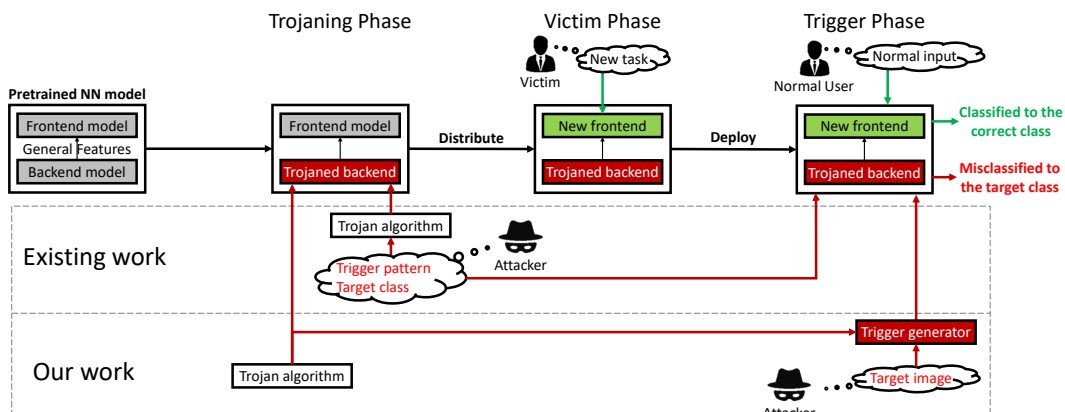

Figure 1: Attack methodology comparison. In existing studies, attackers should decide the trigger pattern and target class pairs in the trojaning phase. In contrast, our method inserts a general trojan in the trojaning phase and decide the target class in the trigger phase.

in the trojaned backend to control the behavior of the system. The modification that can be made highly depends on the scenario of the victim task.

In most real applications, the modification is a small patch in the input image, denoted as *trigger pattern*. The clean input image is called *source image*, and its corresponding label is *source class*. The source image patched with the trigger pattern is denoted as *trigger image*. The corresponding misclassification target is *target class*. It is described by a *target image*.

Note that, although attackers may also perform a black-box adversarial attack in the trigger phase and also lead to misclassification. The sources of the two threats are completely different. Thus, defending adversarial attacks will not reduce the risk of trojaning attacks.

## 4  METHOD

### 4.1  ATTACK METHODOLOGY

The major contribution of our work is the new attack methodology, which greatly extends the power of the trojaning attack.

In the workflow of the existing transfer learning attack shown in Figure 1, attackers choose the trigger pattern and the corresponding target class in the trojaning phase and then modify the backend model to recognize the trigger pattern without affecting its behavior for normal inputs. Then, in the trigger phase, attackers will present the trigger pattern in a normal input to trigger the misclassification as the target class. This attack flow has two major drawbacks.

• **Fixed targets.** The target classes are decided in the trojaning phase. Attackers cannot choose targets on demand in the trigger phase.

• **In-scope targets.** In the trojaning phase, the victim task is completely unknown to the attacker. It is difficult to support target classes that are not included in the class set of the original task.

In Table 1, none of the existing studies support out-scope and dynamic target due to these drawbacks.

We propose a new attack methodology as shown in Figure 1. The difference is that in the trojaning phase, we insert a more powerful programmable trojan and create a corresponding trojan generator. In the trigger phase, we use a *target image* to indicate the target class. The generator will encode the target image into a trigger pattern. It will be presented in the input image and the trojaned backend can decode the trigger pattern and misclassify the input as the target class defined by the target image. The proposed attack methodology solves the two drawbacks due to the following designs.

• **Select targets in the trigger phase.** We insert a general trojan in the clean model and select the target class later in the trigger phase.

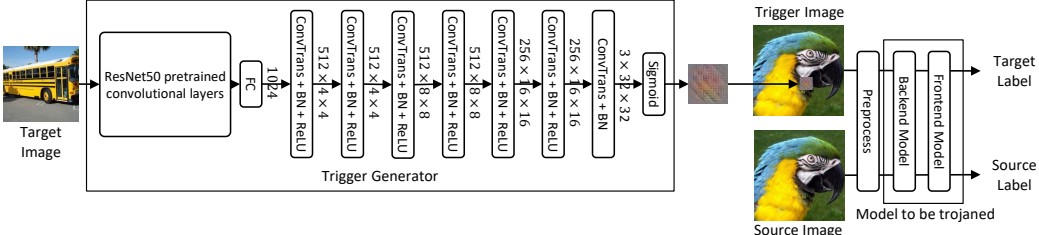

Figure 2: Trojan Insertion. We train a trigger generator together with the front-end model. The left side shows the network architecture of our generator. It accepts a target image and uses ResNet50 pre-trained convolutional layers to encode it into a 1024-length vector. Then we use multiple transposed convolutional layers to generate the trigger pattern from the vector. The trigger pattern will replace part of the source image to form the trigger image. Then it will be fed into the model to be trojaned and trained to predict the target label. To keep the original functionality. Normal source images will also be fed into the model and trained to predict the source label.

• **Describe targets with target images.** We use a target image instead of a target class to describe the intended behavior. It can support any target class on demand.

Moreover, in the trigger phase, attackers may be still unaware of the explicit class sets of the victim model, while the expected behavior of the application system is known to attackers and the victim task is just a sub-task of the application system. Accordingly, with the target image, attackers can program the expected behavior of the application system directly without the information of the explicit classes of the victim task.

### 4.2 Trojan Insertion

During the trojaning phase, the attacker will train the generator and the original model to insert a trojan in the model. We show the expected functionality of the trojaned model and its trigger generator in Figure 2. The trigger generate receives a target image as input and generates a small trigger pattern. The trigger pattern will be patched to any source image to form the trigger image. Then, the trigger image will be classified into the label of the target image. Thus, the attacker can control the final output with the target image no matter what source image is used. Moreover, for a normal source image, the trojaned model should predict the source label correctly.

Formally, the generator $g$ will produce a trigger pattern $z = g(x_{target})$ from a target image $x_{target}$. Then, the trigger pattern $z$ will be patched to a source image $x_{source}$ to form a trigger image $x_{trigger} = P(x_{source}, z)$, where $P$ is a function to patch $z$ in a random position of $x_{source}$. Note that, $P$ is differentiable: its gradients only need to propagate to the region of the trigger pattern directly. Then, we expect the model to predict target label $y_{target}$ when input is $x_{trigger}$ and predict source label $y_{source}$ when input is $x_{source}$. To train the generator and the trojan, we optimize the two functionalities together. The loss function should be as Formula 1, where $L$ is the cross-entropy loss, $\alpha$ is a hyper-parameter to control the weights of normal behavior and trojan behavior and $f$ is the trojaned model.

$$\alpha L[f(x_{trigger}), y_{target}] + (1 - \alpha)L[f(x_{source}), y_{source}] \tag{1}$$

Note that, attackers neither know the victim model nor have access to the victim's dataset. Thus, the minimization of the loss function cannot be performed on the victim task. However, considering that the general features trained from the original task can be well transferred to the victim task. It is also feasible for the attacker to train a general trojan with original tasks, as well. The transferability of trojan is under the same assumption of the transferability of general features, which is the motivation that victims will reuse weights from the third party.

We use the stochastic gradient descent (SGD) algorithm to optimize the parameters of $g$ and the parameters in the backend of $f$. The frontend of $f$ is fixed during the optimization such that only the backend learns the trojan functionality. When the victim train a new frontend, the trojan in the backend can still be effective. For convolutional neural networks (CNNs), the backend is typically the convolutional layers. $f$ is initialized with the clean model and $g$ is initialized randomly. In the loss function, the gradients to $g$ are multiplied with $\alpha$, which is a very small number. Thus, we scale

Table 2: The accuracy of clean and trojaned model and the attack success rate on ImageNet models.

| Model | Clean Model Accuracy top1 / top5 | Trojaned Model Accuracy top1 / top5 | Attack Success Rate top1 / top5 |
|---|---|---|---|
| VGG16 | 73.37%/91.50% | 72.37%/90.96% | 50.27%/75.87% |
| ResNet50 | 76.15%/92.87% | 73.88%/91.66% | 37.55%/65.34% |
| MobileNet-V2 | 71.81%/90.42% | 69.32%/89.14% | 31.04%/57.64% |

the gradient of parameters in $g$ by $1/\alpha$ to have a balanced update between the generator and the trojan.

Figure 2 also shows the network architecture of the generator we used. We first use the convolutional layers of the pre-trained ResNet50 (He et al., 2016) and an FC layer to encode the target image into an internal feature vector of length 1024. Then, we use several transposed convolutional layers to generate a $32 \times 32$ trigger pattern, which is the typical network architecture for image generation in generative adversarial networks (GANs) (Radford et al., 2015). Specifically, we use a sigmoid function in the last layer to produce pixel values between 0 and 1, and then scale each pixel to the interval between 0 and 255. It will be further normalized with the mean and variance values of the ImageNet dataset, which is a typical pre-processing step for ImageNet models. Finally, we patch the trigger pattern in a random position in the source image and feed it into the model to be trojaned. The backend part is its convolutional layers and the frontend part is its FC layers. Note that, during the training, we fix the parameters of the frontend model and the pre-trained ResNet50 in the trigger generator.

### 4.3 TROJAN TRIGGERING

During the triggering phase, the attacker just picks a target image that contains the scenario that he expects the victim's system to see and use it to generate the small trigger pattern. Then, he just presents the trigger pattern in any small region in the input of the victim's system. The victim's system will predict the label of the target image and react as seeing the target image.

## 5 EXPERIMENT AND RESULT

### 5.1 OUTSOURCED TRAINING ATTACK EFFECTIVENESS

We first demonstrate the outsourced training attack on ImageNet models to show the properties of the trojan without victims' modifications. Note that, our trojaning attack is different from existing literature that backdoors a specific pattern for a specific class. Our trojan can support all classes simultaneously in one trojaned model. Thus, we use the averaged success rate for all pairs of the 1000 source classes and the 1000 target classes to measure the capability of our trojan. Existing literature only support one class each time, thus they cannot compare with each other. Moreover, the attack success rate cannot exceed the image recognition accuracy. Otherwise, the generator together with the trojaned model forms a more powerful image recognition model that classify the target image to target label.

**Setup.** We implement the trojan insertion method with PyTorch. We choose VGG16 (Simonyan & Zisserman, 2014), ResNet50 (He et al., 2016), and MobileNet-V2 (Sandler et al., 2018) as the model to be trojaned. Initially, we set $\alpha$ to $10^{-3}$ and choose $10^{-3}$ as the learning rate for all cases. Then, we decrease the learning rate by $10\times$ every 10 epochs. After the loss function converge, we change $\alpha$ to $10^{-4}$, restore the learning rate of target model to $10^{-4}$ and fine-tune the generators and trojans, which enables higher accuracy for the cases of VGG16 and ResNet50. MobileNet-V2 is slightly different: $\alpha$ is set to $5 \times 10^{-4}$ at the fine-tuning phase.

**Results.** In the first step, we evaluate our trojaning attack on VGG16, ResNet50, and MobileNet-V2 in the outsourced training attack scenario. Table 2 shows the accuracy of the clean model and the trojaned model. The accuracy drop is within the accuracy variation of these models. Meanwhile, we achieve a high attack success rate. The trojaned VGG16 has a 50.27% attack success rate across 1000 target classes. It is a high attack success rate since it is comparable with the recognition accuracy, 72.38%. The hyperparameter $\alpha$ is important for the tradeoff between maintaining prediction accuracy on normal inputs and increasing attack effectiveness on trigger inputs. We find that

Table 3: Transfer learning attack results. We use the same trojaned VGG16 model to test the transfer attack success rate on two smaller dataset, Flower and Caltech-256. The trojaned model is made with only ImageNet dataset. Smaller datasets are only used to train victim's FC layers.

| Dataset | Clean Model Accuracy | Trojaned Model Accuracy | Attack Success Rate |
|---------|---------------------|------------------------|--------------------|
| Flower Dataset | 91.70% | 91.56% | 38.15% |
| Caltech-256 | 72.80% | 73.37% | 37.63% |

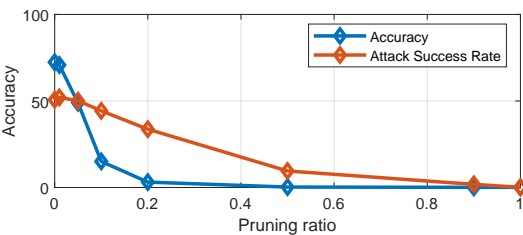

Figure 3: The accuracy vs. attack success rate with different pruning factor for Fine-Pruning (Liu et al., 2018a) defense method.

$\alpha = 10^{-3}$ would be the sweet point. We can further increase the attack success rate by applying a larger $\alpha$, but it will lead to more accuracy drop of the trojaned model.

## 5.2 TRANSFER LEARNING ATTACK EFFECTIVENESS

We also demonstrate an end-to-end transfer learning attack on two small datasets that are independent of ImageNet. The trojaned VGG16 will be fine-tuned for the small datasets. And we test the effectiveness of the trojan after the fine-tuning. No further trojaning modification is made to the trojaned model, we use the trojaned model from the previous section directly. None of the existing trojaning attack methods can be applied to this scenario.

**Setup.** We follow the scenario described in the official tutorials of Tensorflow and MXNet. We use the trojaned VGG16 as an example and train new classifiers with new FC layers for the Flower dataset and Caltech-256 dataset. Finally, we pick two random images from the validation set, one as source image and one as target image, to form a trigger image. We have eliminate the cases that source image and target image are from the same class.

**Result.** In Table 3, We show the transferability of our trojaned model. We use the trojaned VGG16 model mentioned in Table 2 to test its effectiveness on two smaller datasets, the Flower dataset and the Caltech-256 dataset. The two datasets are unknown when trojaning the VGG16 model and the classes in these datasets are completely different from the 1000 classes in ImageNet. Thus, non of existing studies can attack the victims successfully in this case because their misclassification target must be one of the 1000 classes of ImageNet. Our trojaned model can still achieve about 38% success rate on both datasets. Although the absolute value of the attack success rate is not that high, the damage of this attack is still quite severer. One trojaned ImageNet pretrained model can affect almost all models that reuse its convolutional layers.

## 5.3 DEFENSE ANALYSIS

Our goal is to extend the capability of trojaning attack. It also leads to better stealthiness because we train a general trojan instead of simple trojans for certain handcraft patterns and it shows no biased behavior for different target classes.

Defense methods just make initial steps on the simple trojaning attack method with one or a few fixed target classes on small datasets. Chen et al. (2018) detects if the dataset is poisoned, which cannot be applied to our threat model because the trojaned model is trained by the attacker. NeuralCleanse (Wang et al., 2019) detects the trojan based on the biased behavior of the fixed target classes. They assume that only one or minority of fixed classes can be target classes. However, our trojan supports dynamic target class and can generally trigger all the classes; thus, there is no

such bias in our trojan. Fine-Pruning (Liu et al., 2018a) prunes the model using the validation set to reduce the redundancies in order to squeeze the trojan functionality. We test Fine-Pruning on our trojaned VGG16 for ImageNet dataset. The result is shown in Figure 3: with different pruning ratio, the attack success rate dropped as well as the accuracy. Namely, the trojan functionality is highly coupled with the original task; removing trojan will also destroy the well-learned feature as well. The trojan is even more robust than the well-learned features.

The major difficulty of defending the proposed attack is that the trojan shows no biased behavior for all target classes and its functionality is highly coupled with the well-learned features. Moreover, the trigger generator is also a neural network, which can add additional regularization terms to make the trigger more robust and hard to detect. Developing defense methods for this kind of trojaning attack is still challenging.

# 6 Discussion

## 6.1 Variants

The key idea of the programmable trojan is to use an NN to generate the trigger image and train the generator network together with the trojan. Our demonstration is a simple case study. It can be extended to many variants.

**Trigger format.** In our case, we use a small trigger pattern patched in a random position of the source image. Such patterns may be obvious for human, but in some case that the victim's application is using a camera to capture images and process them automatically with the victim model. So, the attacker can easily display a small trigger pattern to trigger the subsequent consequences, such as authorizing the attacker to enter a secure place or misleading a self-driving car into an accident. In some other cases that the attacker could modify the entire image and the modification is imperceivable for humans, like adversarial attacks, we can design the generator to produce the modified full-size input image. Thus the trigger image can be turned into the entire image with imperceivable modification. We can also use the generator to encode the target image into the imperceivable modifications and train the trojan to recognize and decode information from it.

**Trojan capability.** By defining the forward pass of the trigger case, we can make the trojan more robust. In our demonstration, we place the trigger pattern in a random position of the source image to make the trojan robust to the position of triggers. It is also possible to apply other random transformations, such as scale, rotation, to enable the trojan more robust. It is also possible to let the trojan support multiple trigger formats by feeding trigger images from different generator networks. These variants may greatly enhance the threat in the real world.

**Model capacity.** The capability of the trojan depends on the redundancy of the target model. In our demonstration, the network architecture of the trojaned model is fixed and the trojan can only exist in the weight parameters. However, the emerging AutoML (Zoph & Le, 2017) technology enables the algorithm to search the best network architecture for a certain task to maximize accuracy. The obtained network architectures from AutoML algorithms are usually complicated and hard to explain, which further increase the threat of our programmable trojaning attack. The trojan can also be hidden in the network architecture in this case. Attackers can search the best architecture and parameters to maximize the capability of the trojan and publish the pre-trained architecture and parameters online.

# 7 Conclusion

We propose a powerful NN trojaning attack under more practical scenarios. Compared to existing NN trojaning methods, our trojan supports dynamic and out scope target classes, which make it broadly applicable. The trojan can be inserted into large-scale models, which provides well-learned general features. Thus, the trojan can affect a large scope of applications. Further analyses show that the proposed trojaning attack is difficult to be detected or removed for existing defense methods.

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
