# OpenReview forum: "Programmable Neural Network Trojan for Pre-trained Feature Extractor"
_ICLR.cc/2020/Conference — Reject_

### Official Review · AnonReviewer2 · 2019-10-23
**Official Blind Review #2**

**Rating:** 3

**Review:**

This paper proposes an adaption of existing backdoor attacks, with the main goal of enabling backdoor attacks in the transfer learning setting. Specifically, instead of pre-defining the trigger pattern and the target label, they train a neural network to generate different trigger patterns for different target images, so that after the source image is blended with the generated trigger pattern, it will be classified in the same way as the target image. This formulation makes it possible to  generate backdoor attacks that stay effective in the transfer learning setting, when the label set of the fine-tuning task is different from the original task. They evaluate their approach using pre-trained models on ImageNet, and also show transfer learning results using two smaller datasets.

I think studying the effectiveness of backdoor attacks under the transfer learning setting is a good topic. However, I am not convinced that the proposed approach is necessary a good way to do so, and have the following questions:

1. To train the trigger generator, do the authors only train it on the pre-trained dataset, or the images of the downstream task is also used? If training does not use the images from the downstream task at all, then it is interesting that the generator can generalize well, which may suggest that although the downstream task has a different prediction goal, the input images themselves share some similarity to the pre-trained task. Could the authors provide some explanation on it?

2. For the attack success rate, I would like to see more analysis on how the choice of the source and target images affect the attack performance. Specifically, if the source and target images have the same label on the pre-trained task, but different on the downstream task, is it easier or harder to generate successful backdoors for the downstream task? Similarly, what if the source and target images have different labels on the pre-trained task, but the same label on the downstream task?

3. Is it necessary to generate different trigger patterns for every different target image? Is it possible to use the same trigger pattern for multiple target images, at least in the case when they look similar? In general, backdoor attacks would expect that the same trigger pattern can be re-used among different input instances.

-------------
Post-rebuttal comments

Thanks for your response! However, I still think the evaluation is weak, given that the attack success rate is low, and the current version of the paper does not provide a good justification for both the design choices and the empirical results. Therefore, I keep my original assessment.
------------

**Experience Assessment:**

I have published in this field for several years.

**Review Assessment: Checking Correctness Of Derivations And Theory:**

I carefully checked the derivations and theory.

**Review Assessment: Checking Correctness Of Experiments:**

I carefully checked the experiments.

**Review Assessment: Thoroughness In Paper Reading:**

I read the paper thoroughly.

---

> ### Author Response · Authors · 2019-11-06
> **Response to AnonReviewer2**
>
> Thank you for the detailed review.
>
> 1.Generalization
> To train the trigger generator and trojan we only use the dataset for pre-training (i.e. ImageNet). The downstream task (i.e. Flower/Caltech-256) is unknown. The reason for the generalization is that the datasets for pre-training are usually large and general for a large domain (e.g. ImageNet for CV). The ImageNet models can provide general features for the downstream tasks from different subdomains. Thus, it is also reasonable that our generator and trojan trained on ImageNet can also learn a general triggering pattern and pattern extraction method and generalize for those downstream tasks.
> Moreover, this is also the fundamental assumption of the NN trojaning attack. Victims expected to reuse the general features from the models, which gives the attacker an opportunity to hide general trojans together with the general features. The scope of downstream tasks is usually the cases that the dataset for pre-training can transfer well. Thus they are also suitable for the trojan to generalize well.
>
> 2.Source image vs. Target image
> I do not understand the question well. Did you mean that if the target image and source images belong to the same class in the original dataset, but belong to different classes in the downstream tasks, is the attack success rate higher or lower?
> It is difficult to directly analyze this because we do not find a suitable small dataset that provides more detailed categories than ImageNet in a subdomain.
> However, by analyzing the performance of our trojaned model on the Flower dataset, we think there is no obvious correlation between them. The Flower dataset has a class, "daisy", which is the only one appeared in ImageNet. The attack success rate to misclassified to this class is in the middle of all classes. The normal accuracy of each class also shows no difference between all classes. This means the same class as in the original dataset and the downstream dataset does not provide a better transferability than other classes.
> Moreover, as introduced in the answer to your comment-1, the transferability of trojan is under a similar assumption as the transferability of general features. The downstream tasks can usually transfer well for both in-scope class and out-scope class.
>
> 3.Trigger pattern generation
> The target image is used to describe the target class that the attacker expects the victim model to predict. The input is the source image with a small region replaced by the generated trigger pattern.
> Thus, if the attacker expects the victim model to predict to a certain class. He only needs to randomly pick one image (target image) that contains the object of that class and use it to generate the small trigger pattern. Then, the trigger pattern is universal for different input instances (source image).
> The only case that the attack needs to generate a new trigger pattern is that when he expects to switch the target label. For one target label, one trigger pattern is universal for all input instances and even different victim models that contain this class.

---

### Official Review · AnonReviewer3 · 2019-10-23
**Official Blind Review #3**

**Rating:** 6

**Review:**

This paper proposes a general framework for constructing Trojan/Backdoor attacks on deep neural networks, specifically in cases where the end user plans to perform transfer learning on the backdoored classifier. To accomplish this, this work proposes the use of a trigger generator that is trained alongside the trojaned network, that allows the attacker to cause any image to be misclassified as any other. The proposed method is general, and is shown to work across a variety of datasets.

Although the threat model here is slightly less general than the standard backdoor attacks threat model (in that the adversary here also needs to be involved in training the model, and not just supplying the data), the authors do well to motivate the threat model, and the attack is sufficiently general to be an interesting security model. The method is, to the best of my knowledge, novel, and is an innovative way to launch trigger attacks even when the adversary does not know what the final classes or task will be. The evaluation is thorough enough, though the results themselves could be better. However, as this is the first attack of this kind, the results seem sufficient to warrant publication.

My major concern with this work is readability---many of the sections are in need of significant proofreading, and there are many grammatical/word choice mistakes that make the paper somewhat difficult to read. (For example, "P is derivative" in Section 4.2 should probably say "P is differentiable"; the use of frontend/backend is somewhat counterintuitive, as the backend should be the what is given to victims, and the victim trains a frontend reliant on the backend). The method itself could also be presented more clearly (section 4.2 specifically could use less equations and more exposition). Overall the paper requires significant written revision in order to be up to the standard of publication, but given that these concerns can be addressed in the revision period I am (weakly) recommending acceptance.

**Experience Assessment:**

I have published in this field for several years.

**Review Assessment: Checking Correctness Of Derivations And Theory:**

N/A

**Review Assessment: Checking Correctness Of Experiments:**

I carefully checked the experiments.

**Review Assessment: Thoroughness In Paper Reading:**

I read the paper thoroughly.

---

> ### Author Response · Authors · 2019-11-06
> **Response to AnonReviewer3**
>
> Thank you for the detailed review. We appreciate your comments on the contributions of our work and your valuable suggestions to improve our paper.
>
> We have followed your suggestions to proofread and improve the readability of this paper. The revision mainly includes the following modifications:
> - Grammatical mistakes for all sections.
> - We update section 4.2 with more exposition and fewer equations to present our trojan insertion method more clearly.
> - We add section 4.3 to briefly introduce how to trigger the trojan.
> - We switch the use of backend and frontend in both texts and figures.
> - We also add more paragraphs in section 5.1 and section 5.2 to describe our experiment more clearly.
>
> We hope the revised version presents our method more clearly.

---

### Official Review · AnonReviewer1 · 2019-10-24
**Official Blind Review #1**

**Rating:** 1

**Review:**

This paper proposed a "learnable" trojan by training a neural network that takes a sample (e.g. an image) as an input and generates a programmable trigger pattern as an output. The authors argue that the proposed method can support dynamic and out scope target classes, which are particularly applicable to backdoor attacks in the transfer learning setting. The authors conducted experiments on large-scale models (ImageNet) in two settings: (1) outsourced training attack;  (2) transfer learning attack.

Although the idea of making a backdoor attack more robust (e.g. more transferrable) and programmable is interesting, I don't think the current results fully substantiate the claimed benefits. Below are my concerns:

1. Lack of performance comparison: in the outsourced training attack, the attack success rate is quite low ( the best top-1 attack rate is ~50% on VGG). In the existing literature, the backdoor attack success rate can be made nearly 100%. This makes me wonder whether the low attack success rate only occurs in the proposed attacking method. Since there are no results from existing attacks, there is no way to evaluate how good the proposed attack is.

2. How about small dataset/model? I understand that the authors want to emphasize the scalability of their attack to large models like ImageNet. However, there are no comparisons on ImageNet (my comment 1). The authors are suggested to compare performance on standard datasets in backdoor attack literature (e.g. CIFAR-10, traffic sign)

3. It was unclear how "dynamic" the proposed method can be. Based on the attack formulation in equation (3), in order to make the attack "dynamic" in terms of changing different target classes, the attackers need to train a neural trojan network for every target class, which does not seem to be dynamic to me. Can the authors further justify the advantage of the dynamic feature in the proposed attack? And I have concerns about how many target classes an attacker can "dynamically" change. Some experiments showing the number of target classes vs attack performance and clean data accuracy will be very helpful.

4. The defense argument against detection methods is weak. Unless the authors can show the proposed attack has the ability to simultaneously backdoor all possible target classes, simply arguing the attack is dynamic and thus can evade detection is not convincing, not to mention in the backdoor setting, attacker should make the first move before the defender takes action.

*** Post-rebuttal comments
I thank the authors for the clarification. However, I feel my comments have not been fully addressed, especially on the part on justifying 50% attack success rate on VGG should be considered as significant in the considered setup. Without any valid comparisons, I find it difficult to assess the contributions. In addition, the authors did not add new empirical evidence regarding my questions but mainly re-iterated the applicability of the proposed method, so I will remain my review rating.
***

**Experience Assessment:**

I have published one or two papers in this area.

**Review Assessment: Checking Correctness Of Derivations And Theory:**

I carefully checked the derivations and theory.

**Review Assessment: Checking Correctness Of Experiments:**

I carefully checked the experiments.

**Review Assessment: Thoroughness In Paper Reading:**

I read the paper thoroughly.

---

> ### Author Response · Authors · 2019-11-06
> **Response to AnonReviewer1 - Clearify some misunderstandings**
>
> Thank you for the detailed review. From the review comments, We think there are a few misunderstandings to clarify. We will respond to your concerns one by one.
>
> 1.Comparison with existing work
> Our major contribution is not to improve the attack success rate for the same scenario of existing literature. We develop a trojaning method for a more practical scenario. The downstream tasks that reuse the ImageNet model are usually different from classifying the ImageNet dataset. Our method enables outer-scope targets, which make the trojaning attack effective in this scenario.
> The success rate highly depends on the scenario and metric. Directly comparing the success rate is unfair.
> Specifically, in our scenario, we only make modifications to the VGG16 model once and the target label can be any classes including classes outside the scope of the 1000 ImageNet classes. In contrast, existing literature makes modifications on the VGG16 for only one class and the class should be in-scope.
> If compare with existing studies in our outer-sourced scenario. The attacker has only one chance to modify the VGG16 model, then we test the average attack success rate for all 1000 classes in ImageNet. Our method can achieve a 50% success rate while existing methods can only achieve 0.1% (1/1000) success rate because they can only succeed in one target class even if they can 100% succeed on that target.
> Moreover, in our transfer learning scenario. The victim fine-tunes the model for their downstream tasks such as Flower/Caltech-256 dataset, we can still achieve a 38% success rate while the success rate of existing literature is 0% because these datasets do not include any classes in the ImageNet and they are not capable of supporting this scenario.
>
>
> 2.Comparison with a small dataset
> We understand that the reviewer wants to see comparison between the proposed method and existing literature by changing to small datasets. However, small dataset comparison dose not change the capability gap between our work and existing work as introduced in the answer to Comment-1.
> Moreover, a small dataset scenario is also not practical. The assumption of the trojaning attack is based on the motivation of reusing well-learned features. The common practice is to reuse features from large-scale and general models such as ImageNet models in CV or Bert in NLP for downstream tasks and they are both large models.
>
> 3.Dynamic
> Our method is not to extend existing work that backdoors only one target class to a large but still fixed set of classes. Instead, we generate trigger patterns for any desired classes on demand.
> We only need to modify the VGG16 model once and we can support an infinite number of target classes include classes outside the scope of 1000 ImageNet classes. Our trigger pattern is generated from an image (termed as target image) that describes the target class the attacker intends to. Thus the attacker can generate trigger patterns for any target class.
> The advantage of "dynamic" is that we can generate trigger patterns for outer-scope targets. When the attacker releases the trojaned VGG16 in a public model zoo, the victim's downstream task is unknown to him and the downstream task is usually different from ImageNet classification. Our method could still dynamically generate trigger patterns to trigger classes in the downstream tasks.
>
> 4.Defense
> Our attack method can simultaneously backdoor all possible target classes (actually, an infinite number of target classes). All the attack success rates reported in the paper are for all possible target classes. Thus, the defense method based on the biased behavior of the target class will not work. For other defense methods based on reducing redundancy, we also provide Figure-3 to show that it will affect accuracy more severely than attack success rate.
> Moreover, existing defense methods are designed for the existing trojaning methods that target one target class. We are making initial steps on a more practical scenario that differs from existing literature.
>
> We have uploaded a revised version to describe our method more clear. In section 4.2, we add more expositions to explain how our trojan works and we also add section 4.3 to describe how our trojan is triggered. In section 5.1 and section 5.2, we also add more texts to explain the different scenario and capability compared with existing literature more clearly.
>
> Thanks again for reviewing our paper. We feel sorry that the paper does not explain the method clearly and we hope our explanation could dismiss your concern.

---

### Decision · Program_Chairs · 2019-12-19

**Decision:**

Reject

**Comment:**

This paper proposes a general framework for constructing Trojan/Backdoor attacks on deep neural networks. The authors argue that the proposed method can support dynamic and out-of-scope target classes, which are particularly applicable to backdoor attacks in the transfer learning setting. This paper has been very carefully discussed. While the idea is interesting and could be of interest to the broader community, all reviewers agree that it lacks of experimental comparison with existing methods for backdoor attacks on benchmark problems. The paper needs to be significantly revised before publication. I encourage the authors to improve this paper and resubmit to future conference.